# Dataset for Image-based Analysis of Mineral Fertilizer Granules

## Abstract

In the context of the mineral fertilizer industry, a crucial sector for global food production, which faces challenges in production efficiency and fast quality control, this work introduces the Mineral Fertilizer Dataset (MFD), a novel annotated segmentation dataset comprising 1,608 images and 125,648 instances of various fertilizer granules with different colors. Addressing the lack of datasets in this field, the MFD supports both semantic and instance segmentation tasks, with segmentation masks that facilitate the computation of the equivalent area diameter of granules. Periodic checks of the area equivalent diameter based on customer specifications are essential to prevent potential defects, such as caking and dustiness, in the produced fertilizer granules. Baseline models based on Feature Pyramid Network (FPN), UNet, and MANet were trained for semantic segmentation, while baseline models based on Mask R-CNN, YOLOv8, YOLOv9, and Mask2Former were trained for instance segmentation. Our experiments demonstrate the efficacy of these models, as well as the robustness of the trained models in identifying fertilizer granules of different colors not included in our dataset, fertilizer granules under 365 nm ultraviolet light, as well as other granular objects such as Polyethylene Terephthalate (PET) pellets, corn, beans, and even pharmaceutical tablets. This dataset, along with its benchmark results on existing semantic and instance segmentation algorithms, aims to facilitate further advancements in computer vision applications for quality control in the fertilizer industry and related sectors.

## 1 Introduction

In the current era of big data and advanced data analysis, many industrial production lines, including those in the mineral fertilizer production, have yet to fully leverage the potential of machine learning methods due to a lack of specialized datasets and hard to get data from those type of production (Yunovidov et al., 2020). Despite being a crucial and rapidly growing sector, mineral fertilizer production faces significant challenges in meeting the rising global demand driven by population growth. As production and consumption levels increase, so do the quality requirements for these products (Ulrich, 2019). Consequently, large-scale facilities are under pressure to enhance production efficiency and control mechanisms to optimize resource utilization and meet consumer expectations.

Various methods are employed to control particle size in the mineral fertilizer industry, including sieve analysis (Besler, 2008; Kimura et al., 2013), laser scattering (Low-Angle Laser Light Scattering (LALLS)) (ISO, 2009; Lilkov et al., 1999), and opto-electronic control methods (Standardization, 2006; Bjørk et al., 2009; Chávez et al., 2015; Wang et al., 2022). However, each method has its own limitations that restrict its optimal application.

Sieve analysis, while providing high accuracy, does not support continuous monitoring of particle size distribution and is significantly influenced by particle shape Besler (2008). LALLS, despite being fundamentally accurate, is limited by the maximum analyzable particle size (up to 3 mm) and cannot assess shape and color parameters Lilkov et al. (1999); ISO (2009).

Opto-electronic control methods are notable for their versatility, utilizing image analysis to estimate a broad spectrum of parameters, including size, shape, and color (Standardization, 2006; Chávez et al., 2015). Furthermore, this method is currently extensively used in the manufacture of mineral

fertilizers. However, the widespread adoption of these methods is hindered by the lack of readily available datasets and the need for specialized equipment.

In response to the rising trend of optical quality control in manufacturing processes, we constructed and evaluated a dataset comprised of images of mineral fertilizer granules. In this work, we provided an overview of existing semantic and instance segmentation techniques, reviewed existing literature on image-based analysis of fertilizer granules, proposed a novel annotated dataset of mineral fertilizer granules designed for segmentation tasks called the Mineral Fertilizer Dataset (MFD), and trained semantic segmentation models (FPN (Lin et al., 2017b), UNet (Ronneberger et al., 2015), MANet (He et al., 2022)) and instance segmentation models (Mask R-CNN (He et al., 2017), Mask2Former (Cheng et al., 2022), YOLOv8 (Jocher et al., 2023), YOLOv9 (Wang et al., 2024)) on the proposed dataset to serve as baseline benchmarks.

## 2 RELATED WORKS

Pixel classification and image segmentation form the foundation of machine vision. Over the years, image segmentation algorithms have evolved from traditional methods such as thresholding (Otsu, 1979), conditional random fields and global classification (Plath et al., 2009), and k-means clustering (Dhanachandra et al., 2015), to more recent deep learning-based methods, which have proven to be significantly more effective for semantic and instance segmentation. Predicted segmentation masks enable the computation of the area equivalent diameter of granules, following the principles of particle size analysis outlined in ISO 13322-1 (International Organization for Standardization, 2014). Periodic checks of the area equivalent diameter based on customer specifications are crucial for preventing potential defects, such as caking and dustiness, in the produced fertilizer granules. Below, we provide an overview of some existing deep learning-based methods for performing semantic and instance segmentation.

**Semantic segmentation**   Semantic segmentation may be described as a process of classifying pixels with semantic labels. A typical advantage semantic segmentation has over instance segmentation is that, it is less computationally expensive, and can be more readily applied in industrial settings especially when the computers are only equipped with a central processing unit (CPU). Semantic segmentation finds application in various sectors including in to inspect belt conveyor idlers (Siami et al., 2024), concrete surface engineering (Hao & Qi, 2022), pedestrian segmentation (Ullah et al., 2018), recognition of navigable areas (Kim et al., 2023), analyzing medical images (Hatamizadeh et al., 2021; Dhamija et al., 2023), and document scanning and optical character recognition (OCR) (Patil et al., 2022) to mention but a few. Deep learning-based semantic segmentation methods maybe grouped into convolutional neural network (CNN) based methods (U-net (Ronneberger et al., 2015), Unet++ (Zhou et al., 2018), FPN (Lin et al., 2017b)), vision transformer based methods (SegFormer (Xie et al., 2021), Swin-Unet (Cao et al., 2021), SegViT (Zhang et al., 2022)), and methods that utilize both transformers and CNNs (Transunet (Chen et al., 2021), MedT (Valanarasu et al., 2021), Transfuse (Zhang et al., 2021b)). In this work, we have performed experiments using three CNN based models: FPN (Lin et al., 2017b), UNet (Ronneberger et al., 2015), and MANet (He et al., 2022). However, to make this method more applicable to the mineral fertilizer industry and bulk material analysis, we isolated individual granules from the overall mask. To achieve this, the contours of the granules in the predicted binary masks were estimated using topological analysis (Suzuki & be, 1985), allowing the instances of each granule to be obtained from the trained semantic segmentation models.

**Instance segmentation**   Instance segmentation involves detecting and drawing masks on each instance of an object of interest in an image. Instance segmentation methods can be classified into three main categories namely: single-stage, dual-stage, and multi-stage. A single-stage instance segmentation method predicts both object masks and class labels without a separate region proposal neural network. Examples of single-stage instance segmentation methods include: Fully convolutional instance segmentation (FCIS) (Li et al., 2017), Instance-sensitive fully convolutional network (InstanceFCN) (Dai et al., 2016), PolarMask (Xie et al., 2020) and You only look at Coefficients (YOLACT) (Bolya et al., 2019; 2022). The dual-stage instance segmentation method involves first proposing regions of interest, followed by predicting the object masks and class labels using different neural networks. A typical example of the dual-stage method is Mask-RCNN (He et al., 2017).

As the name implies, the multi-stage instance segmentation method involves multiple sequential stages of processing, where each stage refines the instance segmentation results iteratively. Typical examples of multi-stage instance segmentation models include: Cascade Mask R-CNN (Cai & Vasconcelos, 2019), and Recurrent neural networks for semantic instance segmentation (RSIS) (Salvador et al., 2017). Besides the aforementioned models, transformer-based models have been utilized for instance segmentation tasks as well such as: SOLQ (Dong et al., 2021), K-Net (Zhang et al., 2021a), Mask2Former (Cheng et al., 2022), OneFormer (Jain et al., 2023), and Mask DINO (Li et al., 2023). Recently, there has been an increase in zero-shot object detection and instance segmentation methods. Some notable models that have been developed include: the segment anything model (SAM) (Kirillov et al., 2023), and fast segment anything model (FastSAM) (Zhao et al., 2023) which is reported as being fifty times faster than the SAM model. However, these zero-shot models are not yet suitable for deployment in industrial tasks because, they are too slow and hard to maintain. In this work, we performed experiments using Mask R-CNN (He et al., 2017), Mask2Former (Cheng et al., 2022), YOLOv8 (Jocher et al., 2023), and YOLOv9 (Wang et al., 2024) for instance segmentation.

**Image-based analysis of fertilizer granules**   To increase production efficiency, the fertilizer industry is leaning more towards using optical control to assess the quality of produced goods (Wang et al., 2022). Quality control in mineral fertilizer production relies on assessing individual granule characteristics like size, area, and color (UNIDO and International Fertilizer Development Center, 1998). This ensures adherence to customer specifications and identifies anomalies in the production process. By inspecting the quality of produced fertilizer granules, possible environmental pollution is also tackled. Yunovidov et al. (2020) explored a robotic system utilizing classical computer vision for online monitoring of granule size. This system captured images using a high-speed camera and employed image processing techniques to estimate size via ellipses. While color analysis was not implemented, its potential was acknowledged. The system's performance was comparable to the industry-standard Camsizer P4 machine. Building upon their previous work (Yunovidov et al., 2020), Yunovidov et al. (2021) expanded the system's capabilities to encompass granule area, color, and sphericity estimation. Software improvements like adaptive equalization and distance separation enhanced image processing. Additionally, a data recording system documented quality analysis results, enabling both process control and data collection. The upgraded system again demonstrated comparable performance to the Camsizer P4 machine.

Based on our research, there are no similar publicly available datasets. Hence, MFD is the first such dataset to be made publicly available to the research community. The closest datasets we found, which are commonly used in related fields, include the Rice Image Dataset (Koklu et al., 2021), the Corn Grain Dataset (Ribeiro, 2015), and a dataset consisting of 409 images of well-sorted and poorly sorted sediment, terrigenous, carbonate, and volcaniclastic sands and gravels, and their mixtures, used to develop the SediNet model (Buscombe, 2020). All three datasets are suitable for image classification tasks but are not designed for semantic or instance segmentation, which are critical for our intended application. These segmentation tasks enable the computation of the area equivalent diameter of produced fertilizer granules, in accordance with the ISO 13322-1 standard (International Organization for Standardization, 2014).

## 3   MINERAL FERTILIZER DATASET

The MFD dataset comprises 1,608 annotated real images of various types of mineral fertilizer granules captured in different fertilizer production plants, encompassing 125,648 instances of fertilizer granules. Figure 1 displays sample images of these mineral fertilizer granules, including Potassium ore (KCl), Ammonium Nitrate ($NH_4NO_3$), and mineral fertilizers containing phosphorus (Diammonium Phosphate (DAPh) and NPK). Static images of sampled 100g of DAPh, and NPK granules were captured using a camera with a rolling shutter at a resolution of 1920 x 1080 pixels. On the other hand, images of the other fertilizer granule types were captured using a camera with a global shutter at a resolution of 1280 x 1024 pixels. Images of $NH_4NO_3$ were captured dynamically on the conveyor belt.

The Computer Vision Annotation Tool (CVAT) (CVAT.ai Corporation, 2023) was used to annotate the images of the fertilizer granules. Figure 2 shows an annotated image within the CVAT platform. Three different individuals annotated the dataset. Initially, a test dataset (educational dataset)

(a) $NH_4NO_3$      (b) NPK      (c) DAPh      (d) KCl

Figure 1: Mineral fertilizer granules

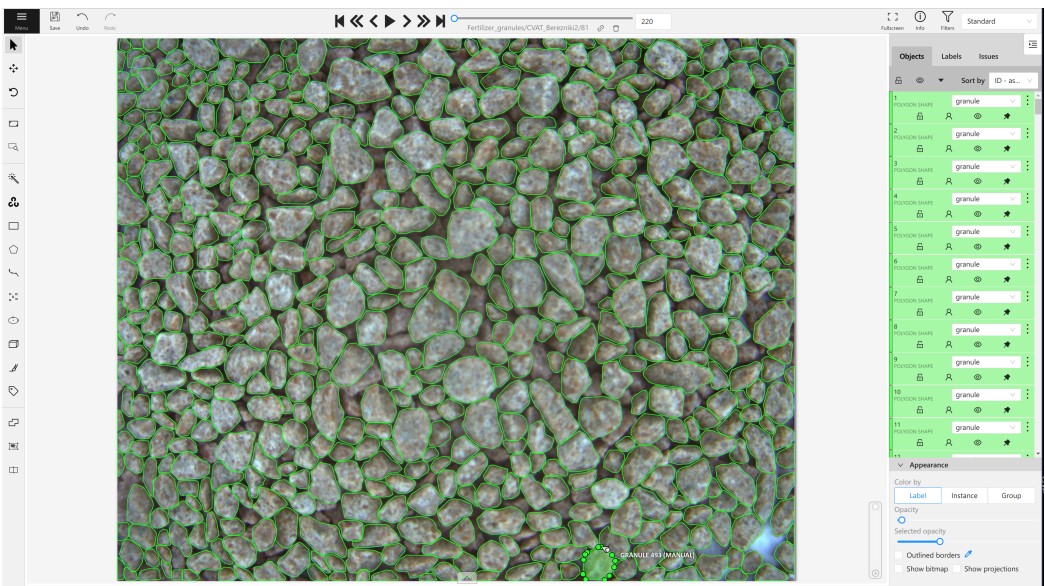

Figure 2: Annotated KCl fertilizer granule in CVAT

consisting of 20 images of fertilizer granules was annotated by these individuals separately. The Intersection over Union (IoU) was calculated among the three sets of annotations for the educational dataset. Only when the IoU exceeded 80% were the annotators permitted to annotate the main dataset. The IoU was calculated using Equation 1, where A and B represent the annotation masks of the annotators.

$$IoU = \frac{n(A \cap B)}{n(A \cup B)} \tag{1}$$

After annotating the images, we filtered the annotation of images with overlapping granules, and several layers to preserve only the first layer of totally visible granules. The images were then split into smaller tiles of 480 x 480 pixels while preserving the annotations and ensuring that each tile was unique using a self developed algorithm. Geometric transformations such as random rotation, random scaling, and cropping were also applied to make our dataset balanced. Additionally, we used multiple iterations of erosion and dilation with a 3x3 pixel elliptical kernel to smooth the masks obtained after manual annotation and dataset balancing. Table 1 provides an overview of the MFD dataset, Figure 3 shows the distribution of granules in the images that make up the dataset, and Figure 4 shows a typical annotated image with the segmentation masks displayed after geometric transformations and before 480 x 480 tiles splitting.

Figure 3: Violin plot showing the mineral fertilizer granule distributions

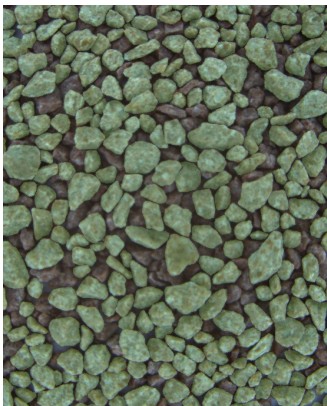

Figure 4: KCl with displayed masks after geometric transformations

## 4 BENCHMARK EXPERIMENTS

The MFD dataset is the first of its kind and would be very valuable for either semantic or instance segmentation to the mineral fertilizer industry as well as industries that work with objects of similar morphology such as pellets, grains, and even pharmaceutical tablets. The models were trained on images of 480 x 480 pixels, for 100 epochs on an NVIDIA RTX A2000 12GB Graphics Processing Unit (GPU). We used 80% of the dataset for training, and 20% for validation.

Table 1: Overview of the MFD Dataset

| Granule Type | Image Count | Instances |
|---|---|---|
| DAPh | 402 | 17063 |
| KCl | 403 | 39165 |
| $NH_4NO_3$ | 402 | 50240 |
| NPK | 401 | 19180 |
| Total | 1608 | 125648 |

Table 2: Performance of semantic segmentation models on the MFD Dataset

| Model | Backbone | mIoU |
|---|---|---|
| FPN | mobilenetv3_large_100 | 0.859 |
| UNet | mobilenetv3_large_100 | 0.869 |
| MANet | mobilenetv3_large_100 | 0.875 |

## 4.1 Semantic Segmentation Using FPN, UNet and MANet

We explored the performance of three semantic segmentation models (FPN (Lin et al., 2017a), UNet (Ronneberger et al., 2015), and MANet (He et al., 2022)) on the mineral fertilizer dataset. For these three models, mobilenet v3 large (Howard et al., 2019) was used as the backbone for feature extraction.

The semantic segmentation experiments were conducted in three stages. First, the binary masks of the fertilizer granules were preprocessed using three iterations of erosion with a 3×3 elliptical kernel to separate granules in the masks that appeared to be joined. Second, the segmentation models were trained using a combination of binary cross-entropy (BCE) (Yi-de et al., 2004), dice (Sudre et al., 2017), and boundary difference over union (Sun et al., 2023) loss functions as shown in Equation 2. Third, based on the predicted binary masks from the trained models, the contours of each granule instance were estimated using topological analysis (Suzuki & be, 1985).

$$\mathcal{L} = 0.2 \cdot \text{BCE} + 0.4 \cdot \text{Dice Loss} + 0.4 \cdot \text{Boundary DoU} \tag{2}$$

Using the predicted binary masks from the trained semantic segmentation models, the contours of the granules were estimated through topological analysis (Suzuki & be, 1985), implemented in OpenCV (Bradski, 2000), allowing the instances of each granule to be obtained. The trained models can operate on a CPU and are suitable for fast assessments. The results obtained using these three semantic segmentation models are summarized in Table 2. Among them, MANet outperformed FPN and UNet.

## 4.2 Instance Segmentation Using Mask R-CNN, YOLOv8, YOLOv9, and Mask2Former

Instance segmentation is a crucial step in analyzing mineral fertilizer granules, as it allows us to identify and isolate individual granules within an image. In this section, we explore the application of Mask R-CNN (He et al., 2017), YOLOv8 (Jocher et al., 2023), YOLOv9 (Wang et al., 2024), and Mask2Former (Cheng et al., 2022) for instance segmentation of mineral fertilizer granules.

The YOLOv8 models were trained on images of 480 x 480 pixels, except the YOLOv8l-seg and the YOLOv9 models which were trained on images of 320 x 320 pixels to accommodate our computing resources.

Table 3 provides a summary of the performance of the trained models. The results show that the YOLOv8 and YOLOv9 models performed better than the Mask R-CNN and Mask2Former models. It is possible to tweak the hyperparameters of these models but we used the default parameters to estimate the baseline performance of these models. The ResNet-50 backbone with Feature Pyramid Network (FPN) was used to train the Mask R-CNN and Mask2Former models. Another keen ob-

Table 3: Performance of models on the MFD Dataset. Models with * where trained using 320 x 320 pixels images.

| Model | $mAP^{box}50$ | $mAP^{box}50-95$ | $mAP^{mask}50$ | $mAP^{mask}50-95$ |
|---|---|---|---|---|
| Mask R-CNN | 0.747 | 0.597 | 0.747 | 0.600 |
| YOLOv8n-seg | 0.939 | 0.759 | 0.927 | 0.675 |
| YOLOv8s-seg | 0.950 | 0.786 | 0.937 | 0.698 |
| YOLOv8m-seg | 0.952 | 0.796 | 0.945 | 0.727 |
| **Mask R-CNN*** | 0.659 | 0.527 | 0.659 | 0.529 |
| **YOLOv8n-seg*** | 0.926 | 0.726 | 0.898 | 0.559 |
| **YOLOv8s-seg*** | 0.940 | 0.763 | 0.913 | 0.588 |
| **YOLOv8m-seg*** | 0.946 | 0.778 | 0.918 | 0.604 |
| **YOLOv8l-seg*** | 0.948 | 0.789 | 0.925 | 0.618 |
| **YOLOv9c-seg*** | 0.947 | 0.778 | 0.918 | 0.602 |
| **YOLOv9e-seg*** | 0.948 | 0.782 | 0.924 | 0.615 |
| **Mask2Former*** | 0.723 | 0.564 | 0.724 | 0.550 |
| Mask2Former | 0.723 | 0.569 | 0.731 | 0.576 |

Table 4: Inference speed of trained models on different devices

| | Frames per Second (FPS) | | |
|---|---|---|---|
| Model | RTX A2000 | RTX 4050 Max-Q | Intel Core™ Ultra 7 |
| MANet | 71.74 | 69.06 | 8.50 |
| Mask R-CNN | 64.90 | 41.98 | 3.19 |
| Mask2Former | 49.21 | 25.75 | 4.56 |
| YOLOv8m-seg | 66.82 | 57.03 | 4.84 |

servation is that the YOLO models that were trained on 480 x 480 pixels images performed better than those trained on 320 x 320 pixels images. Factors such as increased input data, augmented data variety, or the 640-pixel input size for YOLO could contribute to this.

Figures 5, 6, 7, and 8 illustrate the inferences made by the trained models on our test images with a confidence threshold of 0.70. Figures 9, 10, and 11 demonstrate the robustness of the trained models in segmenting fertilizer granules of various colors not included in our dataset, while Figure 12 highlights the models' performance under ultraviolet light. Additionally, Figures 13, 14, 15, and 16 showcase the models' ability to segment objects with similar morphology.

### 4.3 INFERENCE SPEED ON DIFFERENT DEVICES

The inference speed of the trained models was measured on three devices: one with an NVIDIA RTX A2000 12GB graphics processing unit (GPU), another with an NVIDIA GeForce RTX 4050 Max-Q 6GB GPU, and a third with an Intel Core™ Ultra 7 155H Meteor Lake-P central processing unit (CPU) without a GPU. The inference speed was determined by computing the average of the total time required for pre-processing, inference, and post-processing for each image in the test dataset. The results of our experiment are summarized in Table 4. From the table, it is evident that selected models can be used in real-time applications with GPU unit and can be used for periodical control in CPU devices.

## 5 CONCLUSION

We have presented a robust annotated instance segmentation dataset of mineral fertilizer granules with different colors, consisting of 1,608 images and 125,648 instances. This dataset bridges the existing gap of a lack of datasets for instance segmentation in the fertilizer industry and can serve as a baseline for further analysis of the quality of produced fertilizer granules. Additionally, it can be used to develop industrial optical control systems for bulk materials, even in compliance with

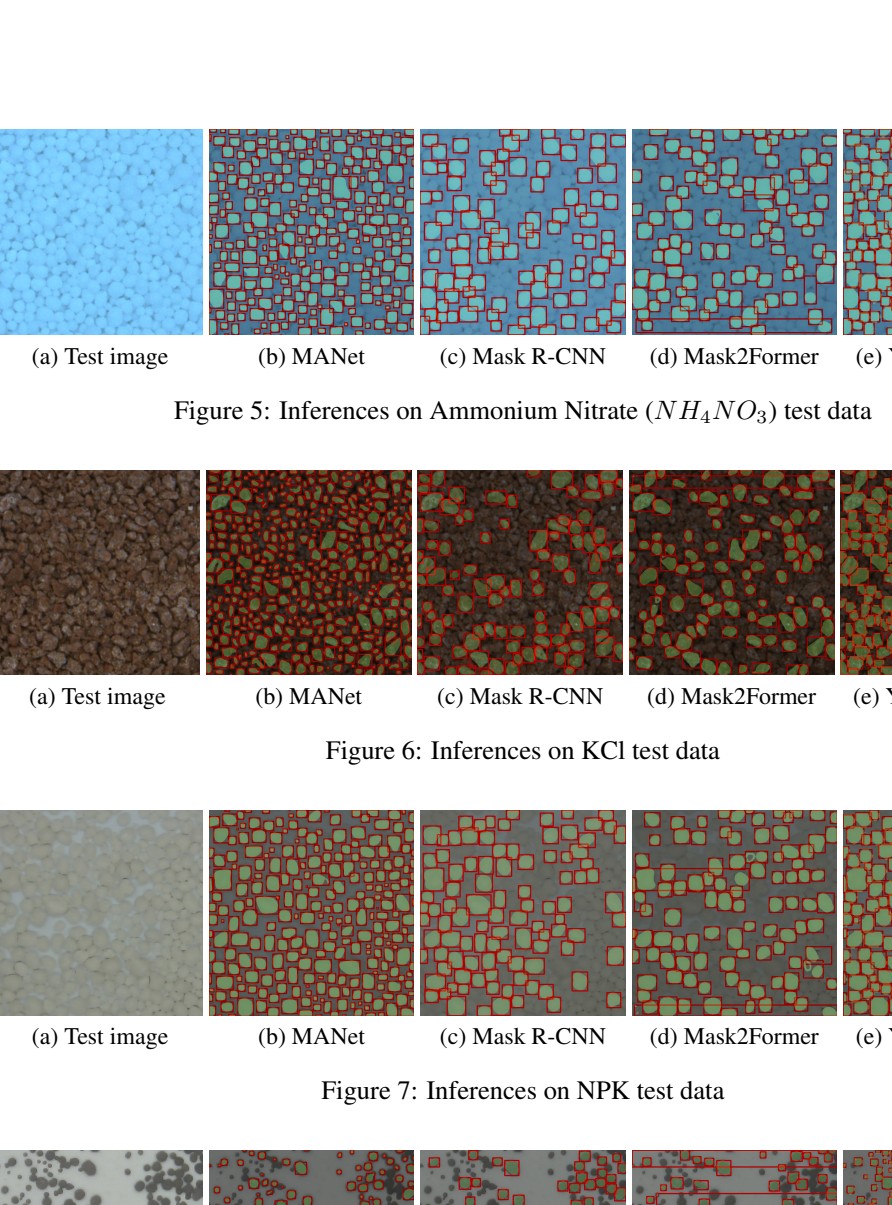

Figure 5: Inferences on Ammonium Nitrate ($NH_4NO_3$) test data

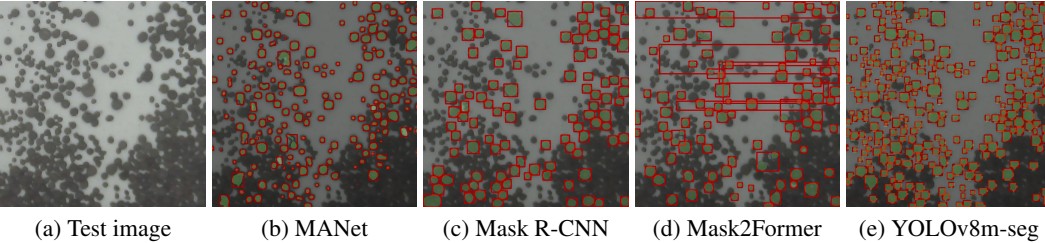

Figure 6: Inferences on KCl test data

Figure 7: Inferences on NPK test data

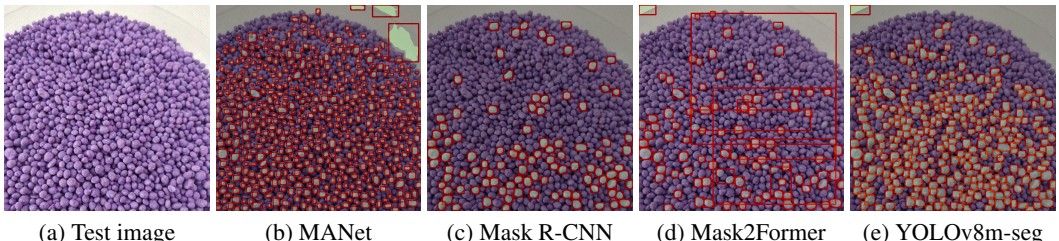

Figure 8: Inferences on DAPh test data

Figure 9: Inferences on purple NPK test data

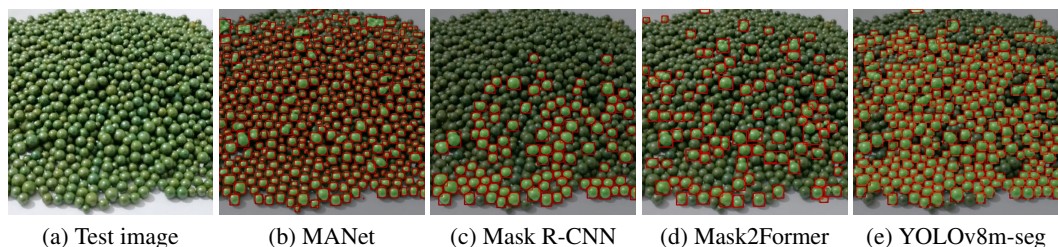

(a) Test image    (b) MANet    (c) Mask R-CNN    (d) Mask2Former    (e) YOLOv8m-seg

Figure 10: Inferences on Amino Acid fertilizer test data

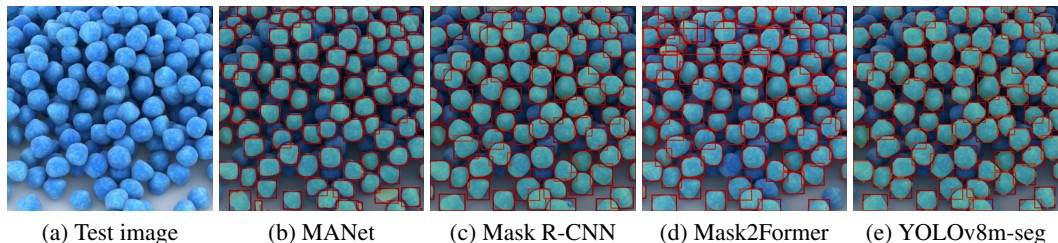

(a) Test image    (b) MANet    (c) Mask R-CNN    (d) Mask2Former    (e) YOLOv8m-seg

Figure 11: Inferences on blue NPK test data

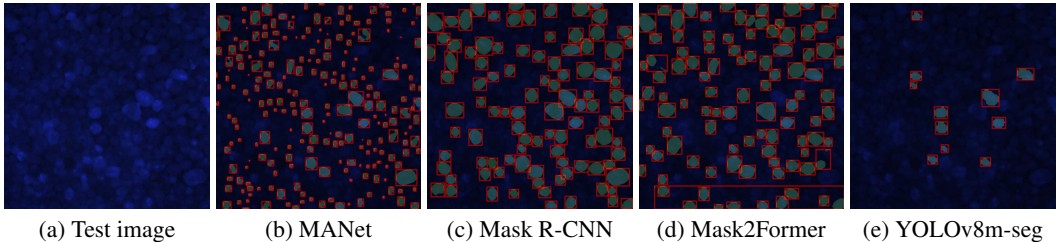

(a) Test image    (b) MANet    (c) Mask R-CNN    (d) Mask2Former    (e) YOLOv8m-seg

Figure 12: Inferences on NPS+B 20-20-14+0.2 oiled fertilizer test data under 365 nm ultraviolet light

ISO 13322-1. We have created a benchmark of established instance segmentation models, including Mask R-CNN, YOLOv8, and YOLOv9. Furthermore, our experiments with fast semantic segmentation models capable of rapid CPU inference show promising results. Combining these models with classical computer vision (CV) post-processing techniques can achieve quality comparable to instance segmentation models for calculating the mask of each granule in an image. We hope that this dataset will pave the way for further advancements in the use of computer vision for quality control purposes in the mineral fertilizer industry.

**Limitations** We considered only the primary fertilizers produced in large-scale continuous processes, which are subsequently used as bases for more complex fertilizers. Additionally, there are many specialized fertilizer blends used in various geographic regions, which we had not test yet. The fertilizer types we have described represent only a small portion of the existing brands and types of such products. We will include more annotated data of DAPh, KCl, $NH_4NO_3$, NPK, and other fertilizer granule types in the MFD dataset to increase its size and variety, which will also enhance the capability of models trained on it. Furthermore, the YOLOv8l-seg, YOLOv9c-seg, and YOLOv9e-seg models were trained on 320 x 320 pixel images due to our currently available computing resources. However, with 640 x 640 pixel images—the default size used to train these models—better performance metrics can be achieved.

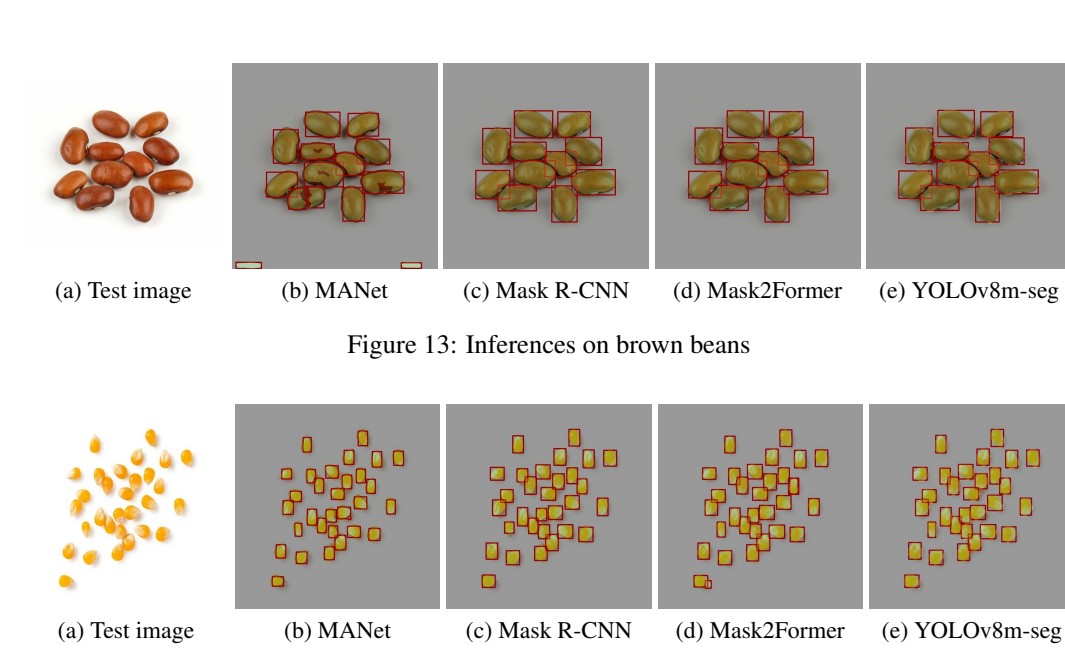

Figure 13: Inferences on brown beans

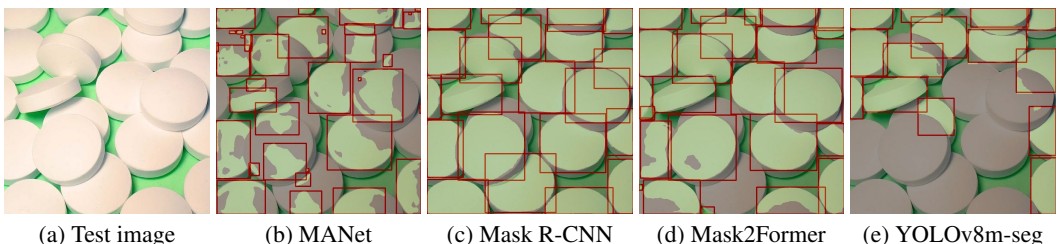

Figure 14: Inferences on corn seeds

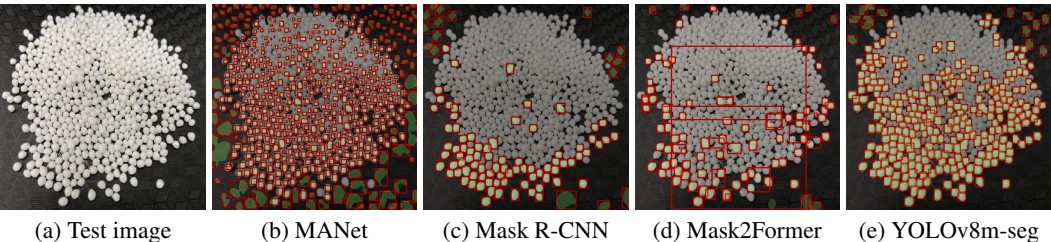

Figure 15: Inferences on pharmaceutical tablets

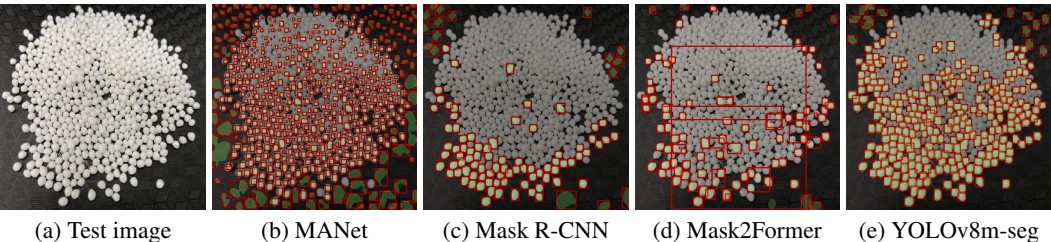

Figure 16: Inferences on Polyethylene Terephthalate (PET) pellets

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

# A APPENDIX

## A.1 DATASET USAGE GUIDE

### A.1.1 EXPERIMENTS WITH YOLO MODELS

The provided dataset is in COCO format. To train the YOLO Models on our dataset, free open source software such as Roboflow platform (Dwyer et al., 2024) can be used to convert the dataset to the required YOLO format, and to split the dataset into training and validation set.

### A.1.2 EXPERIMENTS WITH MASK R-CNN

Mask R-CNN requires data in COCO format; therefore, the dataset can be used as is for experiments with Mask R-CNN. Users may split the dataset into training and validation sets as needed. We used the MMDetection framework (MMDetection Contributors, 2018) to train the Mask R-CNN model.

### A.1.3 EXPERIMENTS WITH MASK2FORMER

Mask2Former requires data in COCO format; therefore, the dataset can be used as is for experiments with Mask2Former. Users may split the dataset into training and validation sets as needed. We used the MMDetection framework (MMDetection Contributors, 2018) to train the Mask2Former model.

### A.1.4 EXPERIMENTS WITH THE SEMANTIC SEGMENTATION MODELS

To use the dataset for experiments with semantic segmentation models, convert the data from COCO format into binary masks using the code below. The segmentation models were trained using PyTorch Lightning (Falcon & The PyTorch Lightning team, 2019) and the Segmentation Models PyTorch package (Iakubovskii, 2019).

```
# Import necessary libraries
import json
import cv2
from tqdm import tqdm
import numpy as np
import os
import matplotlib.pyplot as plt

# Process annotations data
d_path_annot = '../MFD_datasets_coco/annotations/MFD-
    balanced_instances_default.json'
d_path_images = '../MFD_datasets_coco/images'
```

```python
processed_data = {
    'Id': [],
    'image_path': [],
    'semantic_masks': [],
}

with open(d_path_annot, 'r', encoding="utf-8") as json_file:
    json_data_dir = json.load(json_file)
    # Process image data
    for image_inf in tqdm(json_data_dir['images'], desc="Process images: "):
        real_img_id = image_inf['id']
        for k in processed_data:
            processed_data[k].append([])
        processed_data['Id'][-1] = real_img_id
        img_path = os.path.join(
            d_path_images, image_inf['file_name']
        )
        processed_data['image_path'][-1] = str(img_path)
        SIZE = (image_inf['height'], image_inf['width'], 3)
        processed_data['semantic_masks'][-1] = np.zeros(SIZE[:2], dtype=
            np.uint8)

with open(d_path_annot, 'r') as json_file:
    json_data_dir = json.load(json_file)
    image_id_old = ''
    skipped_counter = 1
    morph_kernel = cv2.getStructuringElement(
        cv2.MORPH_ELLIPSE, (3, 3)
    )

    for annotation_data in tqdm(
        json_data_dir['annotations'], desc="Process annotations: "
    ):
        # Data may be numerated to image of have through numeration
        process_image_id = annotation_data['image_id']

        image_data_indx = processed_data['Id'].index(process_image_id)
        label = annotation_data['category_id']
        # Process each granule
        for point_i, point in enumerate(annotation_data['segmentation']):
            if isinstance(point, list):
                point_xy = [
                    [point[j], point[j + 1]] for j in
                    range(0, len(point), 2)
                ]
                cnt = np.array(point_xy).reshape((-1, 1, 2)).astype(
                    np.int32
                )
                if len(cnt) < 3:  # bad contour
                    print('Cnt is bad')
                    continue

                single_mask = np.zeros((480, 480), dtype=np.uint8)
                _ = cv2.drawContours(
                    single_mask,
                    [cnt], -1, label, cv2.FILLED
                )
                single_mask = cv2.morphologyEx(
                    single_mask, cv2.MORPH_ERODE, morph_kernel,
                        iterations=3
                )
                processed_data['semantic_masks'][image_data_indx][
                    single_mask != 0] = label
```

```
                        x, y, w, h = annotation_data["bbox"]
              else:
                  continue

# Visualize Processed Data
image_indx = 0
test_image = cv2.imread(processed_data['image_path'][image_indx])
test_image = cv2.cvtColor(test_image, cv2.COLOR_BGR2RGB)

plt.rcParams['figure.figsize'] = [10, 10]
f, axarr = plt.subplots(1,2)
_ = axarr[0].imshow(test_image, cmap='gray', vmin=0, vmax=255)
_ = axarr[1].imshow(processed_data['semantic_masks'][image_indx], cmap='
    gray', vmin=0, vmax=1)
```

