# OpenReview forum: "Dataset for Image-based Analysis of Mineral Fertilizer Granules"
_ICLR.cc/2025/Conference — Submitted to ICLR 2025_

### Official Review · Reviewer_Saq6 · 2024-11-01

**Soundness:** 2
**Presentation:** 1
**Contribution:** 2
**Rating:** 3
**Confidence:** 5

**Summary:**

The paper introduces the Mineral Fertilizer Dataset (MFD), a novel annotated segmentation dataset containing 1,608 images and 125,648 instances of various fertilizer granules with different colors, aimed at supporting semantic and instance segmentation tasks in the mineral fertilizer industry. The authors trained baseline models such as FPN, UNet, MANet for semantic segmentation, and Mask R-CNN, YOLOv8, YOLOv9, and Mask2Former for instance segmentation, demonstrating the dataset's utility and the models' efficacy in identifying fertilizer granules and other granular objects. The contribution lies in providing a benchmark for computer vision applications in quality control for the fertilizer industry and related sectors.

**Strengths:**

The introduction of the Mineral Fertilizer Dataset (MFD) fills a significant gap in the field by providing a specialized dataset for image-based analysis of fertilizer granules, which was previously lacking.

**Weaknesses:**

The paper's writing is very catastrophic.
The dataset is not very valuable.

**Questions:**

refer weakness.

---

> ### Author Response · Authors · 2024-11-19
> **Response to highlighted weakness**
>
> **Weaknesses**
> 1. The paper's writing is very catastrophic. The dataset is not very valuable.
>
>     - Thank you very much for the feedback. Please explain what is meant by “The paper’s writing is very catastrophic”? In what ways is the paper’s writing catastrophic?
>     - Innovative research is built on high-quality and relevant data. Our work offers a highly valuable dataset to the mineral fertilizer industry. Based on our research and more than 10 years of experience in industrial production of mineral fertilizers, there are no similar publicly available datasets, making MFD the first of its kind to be accessible to the research community. The closest datasets we identified, commonly used in related fields, include the Rice Image Dataset (https://www.kaggle.com/datasets/muratkokludataset/rice-image-dataset), the Corn Grain Dataset (https://www.kaggle.com/datasets/ssrinformatica/2000obj), and a dataset comprising 409 images of well-sorted and poorly sorted sediment, terrigenous, carbonate, and volcaniclastic sands and gravels, along with their mixtures, used to develop the SediNet model (https://github.com/DigitalGrainSize/SediNet). While all three datasets are suitable for image classification tasks, they are not designed for semantic segmentation or instance segmentation of granules in production environments, which are critical for our intended application.

---

### Official Review · Reviewer_Ptqw · 2024-11-03

**Soundness:** 2
**Presentation:** 2
**Contribution:** 2
**Rating:** 3
**Confidence:** 4

**Summary:**

This paper proposes a mineral fertilizer segmentation dataset, featuring real-world scenes with particles of various colors. It makes up for the lack of relevant datasets in the fertilizer industry.And it constructs benchmark test results based on some instance segmentation models and semantic segmentation models, which to a certain extent promotes the development of quality control technology in the fertilizer industry.

**Strengths:**

This paper proposes a mineral fertilizer segmentation dataset, which consists of mineral fertilizer particles of different colors in real scenes, which to a certain extent promotes the development of quality control technology in related industries.

**Weaknesses:**

1. The article lacks innovation. The author only uses some benchmark models for testing on the constructed dataset and does not propose new methods to test on the dataset. The lack of technological innovation also makes the paper look more like an experimental report.
2. In the introduction, the author did not describe the innovation of this paper in points, which is not intuitive enough. In terms of expression, the author did not explain some specific contents clearly, such as how the "self-developed algorithm" is implemented and what are the specific "some OpenCV operations".

**Questions:**

1. The technical innovation is extremely limited. This article only uses some benchmark models for testing on the proposed dataset, and does not make any technical optimization for this field. If there is any technical optimization for this field, it is recommended to add relevant content.
2. For the annotation of the dataset, how does the algorithm developed by the author achieve block uniqueness? What does the author mean by some OpenCV (Bradski, 2000) operations in the article?
3. For the evaluation of the dataset, can you provide more model evaluation effects and some additional indicators to more comprehensively evaluate the quality of the dataset?
4. The article is not well organized in terms of language. It uses conjunctions such as first, second, and at last to describe things in one paragraph, which can easily lead people to mistake it for being AI-generated. Please modify the overall expression of the article.
5. This dataset has only four categories. Can it represent most scenarios in this field?

---

> ### Author Response · Authors · 2024-11-19
> **Response to questions**
>
> **Questions**
> 1. The technical innovation is extremely limited. This article only uses some benchmark models for testing on the proposed dataset, and does not make any technical optimization for this field. If there is any technical optimization for this field, it is recommended to add relevant content.
>
>     - The main focus of this article is to propose a dataset that is the first of its kind in the fertilizer industry. Currently, no such datasets are publicly available to the research community. Hence, our primary technical innovation is the creation of this well-annotated dataset.
>     - Additionally, we trained semantic segmentation models using a combination of Binary Cross Entropy, Dice, and Boundary Difference over Union loss functions to enhance the quality of the predicted segmentation masks. Subsequently, topological analysis implemented in OpenCV was applied to extract the contours of each fertilizer granule instance from the predicted binary masks. The results obtained are comparable to those achieved by instance segmentation models.
>
> 2. For the annotation of the dataset, how does the algorithm developed by the author achieve block uniqueness? What does the author mean by some OpenCV (Bradski, 2000) operations in the article?
>
>     - Thank you very much for the questions. The explanation of the developed algorithm has been updated in Lines 303–308. The OpenCV operation used is topological analysis, which identifies the contours in the predicted segmentation masks.
>
> 3. For the evaluation of the dataset, can you provide more model evaluation effects and some additional indicators to more comprehensively evaluate the quality of the dataset?
>
>     - Using the proposed dataset, the trained models demonstrated promising results, successfully segmenting fertilizer granules of different colors not included in the training dataset, fertilizer granules under 365 nm ultraviolet light, and objects with similar morphology, as shown in Figures 9 through 16.
>
> 4. The article is not well organized in terms of language. It uses conjunctions such as first, second, and at last to describe things in one paragraph, which can easily lead people to mistake it for being AI-generated. Please modify the overall expression of the article.
>
>     - Thank you very much for pointing out this mistake. However, just before the use of "first," "second," and "at last," the final sentence of the previous paragraph states: “The semantic segmentation experiments were done in three stages.” This justifies the use of these conjunctions. To clarify, we have moved this highlighted sentence to the beginning of the next paragraph, preceding the conjunctions "first," "second," and "at last." We have also rephrased the sentences for better clarity, as reflected in Lines 294–299.
>
> 5. This dataset has only four categories. Can it represent most scenarios in this field?
>
>     - As mentioned in the Limitations section, we have considered only the primary fertilizers produced in large-scale continuous processes, which are subsequently used as bases for more complex fertilizers. Additionally, many specialized fertilizer blends are used in various geographic regions, which we have not yet tested. The fertilizer types we have described represent only a small portion of the existing brands and types of such products. Based on our experiments, we have demonstrated that the trained models are capable of segmenting fertilizer types not used in training, as well as objects with similar morphology. This information can be found in Lines 358-359.

---

### Official Review · Reviewer_r52n · 2024-11-03

**Soundness:** 3
**Presentation:** 3
**Contribution:** 3
**Rating:** 6
**Confidence:** 3

**Summary:**

The paper introduces the Mineral Fertilizer Dataset (MFD), a novel annotated segmentation dataset designed for image-based analysis of mineral fertilizer granules. Aiming to address the lack of datasets in the fertilizer industry for improving production efficiency and quality control, MFD includes 1,608 images and 125,648 labeled instances, supporting both semantic and instance segmentation. Baseline models were trained on MFD demonstrate strong efficacy and robustness.

**Strengths:**

1. The Mineral Fertilizer Dataset (MFD) uniquely contributes to the field by explicitly addressing the image-based segmentation of mineral fertilizer granules. This dataset tackles real challenges in the fertilizer industry, where resources for quality control and production efficiency datasets are limited.
2. The paper benchmarks multiple models—FPN, UNet, and MANet for semantic segmentation, and Mask R-CNN, YOLOv8, YOLOv9, and Mask2Former, for instance, segmentation—offering a thorough evaluation across a range of segmentation techniques.
3. The paper provides a detailed overview of the dataset construction process, covering image capture, annotation, and preprocessing steps. It also clearly describes the experimental setup, model selection, and evaluation metrics, ensuring transparency and reproducibility.

**Weaknesses:**

1. This paper does not include a comparative analysis with prior work in this field or similar fields. It would be beneficial to discuss what datasets have been used in previous studies on fertilizer granules or related domains and how the Mineral Fertilizer Dataset (MFD) compares in terms of uniqueness or advantages.
2. This paper lacks a detailed analysis of the dataset's diversity, specifically regarding whether it covers all common types of fertilizer granules and whether these granule types are representative of real-world production.
3. The dataset lacks detailed documentation and user instructions. Information such as the composition of each data element, annotation standards, and a clear breakdown of dataset attributes would make the dataset more accessible and manageable for other researchers.
4. While the paper claims that the dataset and models support real-time and robust applications, it does not provide experimental data to substantiate these claims.

**Questions:**

1. Could the authors provide a comparison between the Mineral Fertilizer Dataset (MFD) and other datasets commonly used in related fields?
2. The paper mentions several types of fertilizer granules but does not specify whether these types cover the full range of granules commonly used in the fertilizer industry. Are there any significant granule types not included in MFD, and if so, how might this affect the dataset’s applicability?
3. Could the authors consider adding a detailed usage guide to help future users better understand and adopt the dataset?
4. Could the authors perform or further discuss tests measuring inference speed across different devices or environments?
5. Could the authors provide experimental results using the same image resolution for all models?
6. Could the authors clarify the criteria for selecting the specific segmentation models in the benchmark?

---

> ### Author Response · Authors · 2024-11-18
> **Response to questions**
>
> **Questions**
>
> 1. Could the authors provide a comparison between the Mineral Fertilizer Dataset (MFD) and other datasets commonly used in related fields?
>
>     - Based on our research, there are no similar publicly available datasets. Hence, MFD is the first such dataset to be made publicly available to the research community. The closest datasets we found, which are commonly used in related fields, include the Rice Image Dataset (https://www.kaggle.com/datasets/muratkokludataset/rice-image-dataset), the Corn Grain Dataset (https://www.kaggle.com/datasets/ssrinformatica/2000obj), and a dataset consisting of 409 images of well-sorted and poorly sorted sediment, terrigenous, carbonate, and volcaniclastic sands and gravels, and their mixtures, used to develop the SediNet model (https://github.com/DigitalGrainSize/SediNet). All three datasets are suitable for image classification tasks, but are not designed for semantic segmentation or instance segmentation of granules in production environments, which are critical for our intended application.
>
> 2. The paper mentions several types of fertilizer granules but does not specify whether these types cover the full range of granules commonly used in the fertilizer industry. Are there any significant granule types not included in MFD, and if so, how might this affect the dataset’s applicability?
>
>     - As mentioned in the Limitations section, we have considered only the primary fertilizers produced in large-scale continuous processes, which are subsequently used as bases for more complex fertilizers. Additionally, many specialized fertilizer blends are used in various geographic regions, which we have not yet tested. The fertilizer types we have described represent only a small portion of the existing brands and types of such products. Based on our experiments, we have demonstrated that the trained models are capable of segmenting fertilizer types not used in training, as well as objects with similar morphology. This information can be found in Lines 358-359.
>
> 3. Could the authors consider adding a detailed usage guide to help future users better understand and adopt the dataset?
>
>     - Thank you very much for this recommendation, we have included a usage guide in the Appendix section.
>
> 4. Could the authors perform or further discuss tests measuring inference speed across different devices or environments?
>
>     - We have included a subsection titled: “Inference Speed on Different Devices” in the updated article. Thank you very much.
>
> 5. Could the authors provide experimental results using the same image resolution for all models?
>
>     - Experimental results using the same image resolution (320 x 320 pixels) for all models have been included in the updated article.
>
> 6. Could the authors clarify the criteria for selecting the specific segmentation models in the benchmark?
>
>     - We selected these segmentation models due to their widespread use across various training frameworks, which facilitates the practical application of our dataset by other researchers. Furthermore, we aimed to explore a diverse range of models, spanning from CNN-based approaches to transformer-based ones.

---

> ### Author Response · Authors · 2024-11-18
> **Response to highlighted weaknesses**
>
> **Response To Highlighted Weaknesses**
>
> 1. This paper does not include a comparative analysis with prior work in this field or similar fields. It would be beneficial to discuss what datasets have been used in previous studies on fertilizer granules or related domains and how the Mineral Fertilizer Dataset (MFD) compares in terms of uniqueness or advantages.
>
>     - Section 2, paragraph 3, discusses prior work in this field. Also, based on our research, there are no similar publicly available datasets. Hence, MFD is the first such dataset to be made publicly available to the research community. The closest datasets we found, which are commonly used in related fields, include the Rice Image Dataset (https://www.kaggle.com/datasets/muratkokludataset/rice-image-dataset), the Corn Grain Dataset (https://www.kaggle.com/datasets/ssrinformatica/2000obj), and a dataset consisting of 409 images of well-sorted and poorly sorted sediment, terrigenous, carbonate, and volcaniclastic sands and gravels, and their mixtures, used to develop the SediNet model (https://github.com/DigitalGrainSize/SediNet). All three datasets are suitable for image classification tasks, but are not designed for semantic segmentation or instance segmentation of granules in production environments, which are critical for our intended application.
>
> 2. This paper lacks a detailed analysis of the dataset's diversity, specifically regarding whether it covers all common types of fertilizer granules and whether these granule types are representative of real-world production.
>
>     - Thank you very much; we have addressed this in the Limitations section.
>
> 3. The dataset lacks detailed documentation and user instructions. Information such as the composition of each data element, annotation standards, and a clear breakdown of dataset attributes would make the dataset more accessible and manageable for other researchers.
>
>     - We have included a dataset usage guide in the appendix.
>
> 4. While the paper claims that the dataset and models support real-time and robust applications, it does not provide experimental data to substantiate these claims.
>
>     - A subsection titled “Inference Speed on Different Devices” has been added to the Benchmark Experiments section.

---

> > ### Comment · Reviewer_r52n · 2024-11-26
> > **Thanks for your feedback!**
> >
> > Thanks for the detailed feedback from the authors. In the first round of reviewing, I raised 4 weaknesses and 6 questions, mainly about the dataset quality and experimental analysis. Currently, the authors' feedback reasonably addresses my concerns. Though some of the other reviewers pointed out that the proposed dataset falls short compared to existing data, I still believe this dataset has novelty and some merits for AI to assist the mineral fertilizer industry. Thus, I still recommend a score of 6, leaning to accept this paper.

---

> > > ### Author Response · Authors · 2024-11-27
> > >
> > > Thank you very much for reviewing our article. We highly appreciate your effort and detailed feedback.

---

### Official Review · Reviewer_WWJx · 2024-11-03

**Soundness:** 3
**Presentation:** 2
**Contribution:** 2
**Rating:** 3
**Confidence:** 3

**Summary:**

This paper presents the Mineral Fertilizer Dataset (MFD), created specifically for the segmentation of fertilizer granules. The dataset contains 1,608 annotated images of four types of mineral fertilizer granules: KCl, NH₄NO₃, DAP, and NPK. The authors assess the performance of classical semantic and instance segmentation techniques using the MFD, highlighting its applicability for fertilizer granule segmentation tasks.

**Strengths:**

The constructed dataset offers a novel approach to evaluating the quality of mineral fertilizer products. The granule annotation process is both logical and compelling. The benchmark results can provide good guidance to researchers in the related fields.

**Weaknesses:**

While the paper provides a benchmark evaluation of several classical semantic and instance segmentation techniques on the MFD, it lacks a significant technical contribution to the field. Given the unique features of the dataset, the authors could consider proposing a tailored model to achieve better segmentation performance.

The authors claim that the MFD dataset is designed for quality control in the fertilizer industry. However, it would be helpful to demonstrate how the segmentation results directly contribute to the quality evaluation process. For example, how do the segmentation metrics (e.g., accuracy, IoU) correlate with key quality control parameters in fertilizer production, such as granule size distribution or shape uniformity? Additional explanations are also needed to clarify the displayed experimental results.

The experiments conducted on extended datasets with similar morphological characteristics lack sufficient detail. A detailed table or description specifying the models used to segment beans, seeds, and tablets, along with their corresponding performance metrics, would strengthen the extended experiments.

Although the dataset may be valuable for the fertilizer industry, the paper lacks a clear discussion of novel ideas or distinguishing contributions. It would be beneficial to explicitly state the novel contributions or to draw a more direct comparison between this dataset and approach and existing methods in the fields of industrial quality control or granular material analysis.

**Questions:**

1. The authors state in the Abstract, "our experiments demonstrate ... the robustness of the trained models in identifying fertilizer granules of different colors not included in our dataset." However, this claim is not supported in the experiment section. Which experiment validates this statement? Given that the segmentation networks used are classical ones, like FPN, Unet, and MANet, what exactly do the authors mean by this claim?

2. In Line 85, the authors mention that isolating individual granules from the overall mask makes the method more acceptable for the fertilizer industry. Why would this isolation process improve industry acceptance?

3. What does the x-axis of the Violin plot in Figure 3 represent for each type of fertilizer granule?

4. In Figure 4, why are only a few KCl granules annotated?

5. What is the intended explanation in Lines 262–268?

6. Which models were employed to segment objects with similar morphology, as shown in Figures 9 through 12?

---

> ### Author Response · Authors · 2024-11-17
> **Response to questions and highlighted weaknesses**
>
> **Questions**
>
> 1) The authors state in the Abstract, "our experiments demonstrate ... the robustness of the trained models in identifying fertilizer granules of different colors not included in our dataset." However, this claim is not supported in the experiment section. Which experiment validates this statement? Given that the segmentation networks used are classical ones, like FPN, Unet, and MANet, what exactly do the authors mean by this claim?
>      - Thank you very much for pointing out this omission. We have included images of fertilizer granules in various colors in the revised article. By robustness of the trained models, we mean that the models are capable of segmenting fertilizer granules of different colors, even those not included in the dataset used for training. Additionally, we validated the robustness of the trained models by evaluating their performance on images captured under 365 nm ultraviolet light, which significantly expands the utility of the proposed dataset for the mineral fertilizer industry. Ultraviolet light is commonly used to analyze the quality of coating-improving additives on granules. Images demonstrating the models’ performance under ultraviolet light have also been added.
>
> 2. In Line 85, the authors mention that isolating individual granules from the overall mask makes the method more acceptable for the fertilizer industry. Why would this isolation process improve industry acceptance?
>     - Isolating the segmented granules is necessary if the model used to detect these granules is a semantic segmentation model. This is because, in semantic segmentation, all granules in a given mask will be regarded as just one object without considering the instances of the separate granules.
>
> 3. What does the x-axis of the Violin plot in Figure 3 represent for each type of fertilizer granule?
>     - The x-axis of the Violin plot in Figure 3 represents each fertilizer type while the y-axis shows the distribution of granules in the images of each fertilizer type.
>
> 4. In Figure 4, why are only a few KCl granules annotated?
>     - In Figure 4, only the granules in the top layer that are fully visible are annotated, which is sufficient for further analysis of the particle size distribution. This is also noted in Lines 207–208 in the updated manuscript.
>
> 5. What is the intended explanation in Lines 262–268?
>     - The intended explanation in Lines 262-268 (lines 293-299 in the updated manuscript) is: The semantic segmentation experiments were conducted in three stages. First, the binary masks of the fertilizer granules were preprocessed using three iterations of erosion with a 3×3 elliptical kernel to separate granules in the masks that appeared to be joined. Second, the segmentation models were trained using a combination of binary cross-entropy (BCE), dice, and boundary difference over union loss functions. Third, based on the predicted binary masks from the trained models, the contours of each granule instance were estimated using topological analysis. We have updated the explanation. Thank you very much.
>
> 6. Which models were employed to segment objects with similar morphology, as shown in Figures 9 through 12?
>     - The models used to segment objects with similar morphology, as shown in Figures 9 through 12, are indicated in the captions under each image. In the updated article, these figures have been renumbered as Figures 13 through 16.
>
>
> **Response To Highlighted Weaknesses**
>
> a. About how the segmentation results directly contribute to the quality evaluation process:
>
>     - The segmentation results contribute to the quality evaluation process by enabling the estimation of the area equivalent diameter of the granules through the predicted masks, which is highly recommended by ISO 13322-1 “Particle size analysis - Image analysis methods”. Periodic checks of the area equivalent diameter based on customer specifications are crucial for preventing potential defects, such as caking and dustiness, in the produced fertilizer granules.
>
> b. A detailed table or description specifying the models used to segment beans, seeds, and tablets, along with their corresponding performance metrics, would strengthen the extended experiments.
>
>     - This information is already included in the article. The models used to segment beans, seeds, and tablets are specified in the captions, and their performance is detailed in Table 3.

---

> > ### Comment · Reviewer_WWJx · 2024-11-22
> >
> > I appreciate the authors' efforts in improving the work and acknowledge the significant amount of effort involved in creating a dataset, which is very important to the research field. However, I do not believe the current work meets the publication standard required for ICLR.

---

> > > ### Author Response · Authors · 2024-11-23
> > >
> > > Thank you very much for the feedback. According to the information in the call for papers for ICLR 2025, under the subject areas, "datasets and benchmarks" is listed and our work is tailored in this direction. Please, is there a metric provided somewhere against which we can compare our work to measure if it meets the publication standard required for ICLR? We would be very grateful if you could share this information with us. Thank you very much.

---

> > > > ### Comment · Reviewer_WWJx · 2024-11-24
> > > >
> > > > Like the research papers, it is hard to define a specific metric for evaluating dataset-related work. You may refer to previously published dataset papers for guidance, such as those presented at CVPR 2024.
> > > > 1. LaMPilot: An Open Benchmark Dataset for Autonomous Driving with Language Model Programs
> > > > 2. SportsHHI: A Dataset for Human-Human Interaction Detection in Sports Videos
> > > > 3. Event Stream-based Visual Object Tracking: A High-Resolution Benchmark Dataset and A Novel Baseline
> > > > 4. 4D-DRESS: A 4D Dataset of Real-World Human Clothing With Semantic Annotations

---

> > > > > ### Author Response · Authors · 2024-11-26
> > > > >
> > > > > Thank you very much for the recommended CVPR 2024 articles. While the suggested CVPR 2024 datasets offer valuable insights and diverse approaches to evaluating dataset-related work, our specific use case presents unique challenges that may not align with their methodologies and evaluation metrics.
> > > > >
> > > > > Here's a brief overview of the unique contributions of the proposed articles:
> > > > >
> > > > > * LaMPilot: This dataset focuses on autonomous driving, integrating language models into driving scenarios. While this is an intriguing area of research, our primary goal is to develop fast and reliable models for industrial applications, specifically those related to ISO 13322-1 "Particle size analysis — Image analysis methods — Part 1: Static image analysis methods". We believe that research in this direction would be more relevant to our industry.
> > > > > * SportsHHI: This dataset is designed for human-human interaction detection in sports videos, emphasizing complex social interactions in dynamic environments. However, their data involves a smaller number of objects and does not require segmentation. Instead, we focus on multilayer, object-rich images to enable precise calculations of size, color, and area equivalent diameters.
> > > > > * Event Stream-based Visual Object Tracking: This dataset addresses object tracking in high-resolution video sequences using event streams. While this is an important area of research, it differs significantly from our main objective. We have not identified any reliable metrics for comparison in this context.
> > > > > * 4D-DRESS: This dataset focuses on real-world human clothing, providing 4D data with semantic annotations. While there are some similarities in our aim to analyze semantic annotations, their dataset involves a single object per image and a vastly different application area and specific needs. We believe that our proposed metrics, benchmarks, and approach provide more suitable information for mineral fertilizer producers to achieve their goals.
> > > > >
> > > > > In conclusion, the environment and data characteristics of our industry diverge significantly from those explored in the mentioned datasets. Therefore, we believe that our proposed metrics, benchmarks, and approach are more suitable for our industry and granular producers. We hope that by considering these factors, you will re-evaluate our work and recognize its potential to address the unique challenges and opportunities within our industry. We highly appreciate your efforts and your opinion.

---

### Meta-Review · Area_Chair_Bxum · 2024-12-19

**Metareview:**

In this work, most reviewers vote for rejection, considering the limited technical novelty and bad writing. After checking the paper, the AC acknowledges the merit of the proposed dataset. However, the paper only benchmarks the existing method and does not propose the method for this specific task. Thus, the contribution is not enough for the current stage.

Furthermore, too many blanks in the paper would affect the reading experience of the paper—for example, the page 4 and 5.

Therefore, the AC tends to reject this work.

**Additional Comments On Reviewer Discussion:**

The author should polish the paper carefully, given the merits of the proposed dataset.

---

### Decision · Program_Chairs · 2025-01-22

Reject